# Circulating Tumor DNA in Head and Neck Squamous Cell Carcinoma

**DOI:** 10.3390/cancers15072051

**Published:** 2023-03-30

**Authors:** Anna Brandt, Benjamin Thiele, Christoph Schultheiß, Eveline Daetwyler, Mascha Binder

**Affiliations:** 1Department of Internal Medicine 5, Hematology and Oncology, University Hospital of Erlangen, 91054 Erlangen, Germany; 2Department of Oncology, Hematology and Bone Marrow Transplantation with Section of Pneumology, University Medical Center Hamburg-Eppendorf, 20251 Hamburg, Germany; 3Internal Medicine IV, Oncology/Hematology, Martin-Luther-University Halle-Wittenberg, Ernst-Grube-Straße 40, 06120 Halle (Saale), Germany; 4Division of Medical Oncology, University Hospital Basel, 4031 Basel, Switzerland

**Keywords:** head and neck squamous cell carcinoma (HNSCC), cell-free DNA (cfDNA), liquid biopsy, monitoring, resistance, prognostication

## Abstract

**Simple Summary:**

Head and Neck Squamous Cell Carcinomas (HNSCCs) are cancers that originate from cells of the head and neck region, including the mouth, nose, and throat. The diversity of these cell types is also mirrored by the high number of different mutations that promote cancer development and progression. For the clinical management of this disease, it is important to identify biomarkers that allow early detection or predict relapse and resistance to therapy. A non-invasive way to monitor these markers over time are so called liquid biopsies, which mostly refers to the detection and analysis of tumor cells or cell-free DNA (cfDNA) in the blood of patients. This review summarizes our current understanding of HNSCC genetics and discusses how the detection of genetic variation in the cfDNA of HNSCC patients can be used to monitor disease and guide therapy.

**Abstract:**

Tumors shed cell-free DNA (cfDNA) into the plasma. “Liquid biopsies” are a diagnostic test to analyze cfDNA in order to detect minimal residual cancer, profile the genomic tumor landscape, and monitor cancers non-invasively over time. This technique may be useful in patients with head and neck squamous cell carcinoma (HNSCC) due to genetic tumor heterogeneity and limitations in imaging sensitivity. However, there are technical challenges that need to be overcome for the widespread use of liquid biopsy in the clinical management of these patients. In this review, we discuss our current understanding of HNSCC genetics and the role of cfDNA genomic analyses as an emerging precision diagnostic tool.

## 1. Introduction

Head and neck squamous cell carcinoma (HNSCC) is a type of cancer that affects the epithelial cells that line regions of the head and neck, including the mouth, nose, and throat (Figure 1) [1,2,3,4]. HNSCC is the seventh most common cancer worldwide, accounting for 3% of all new cancers and for 1.5% of all cancer deaths according to global cancer statistics (GLOBOCAN) [5]. Risk factors for HNSCC include tobacco and alcohol use, viral infections (human papillomavirus [HPV], Epstein-Barr virus [EBV]), poor oral hygiene, exposure to certain chemicals as well as some genetic syndromes, such as Fanconi anemia [1,2,3,4,6]. The majority of patients are diagnosed with locoregional disease, without evidence of metastatic spread, which is either amenable to surgery or chemoradiation (CRT) [1,4,7]. In patients with intermediate to high-risk resectable tumors, adjuvant treatment, consisting of radiotherapy or CRT, is used to reduce the risk of recurrence and to improve outcomes [1,8,9,10]. Despite advances in treatment, including adapting the chemotherapy and radiotherapy protocol, the recurrence rate remains high. Disease recurrence or development of metastases is reported in 50–60% of patients [4]. Whereby, many HNSCC relapses are locoregional and if detected early, can still be treated successfully [11,12,13]. Therefore, early relapse detection is a priority in the management of HNSCC, and efforts should be made to improve surveillance and monitoring strategies to detect recurrence as soon as possible. At present, these strategies involve regular imaging, clinical examinations, and biopsy.

The identification of non-invasive, blood or saliva-based biomarkers to detect HNSCC relapse would be a significant advance in the management of this disease. Cell-free deoxyribonucleic acid (cfDNA), is a promising candidate for such a biomarker, as it has been shown to reflect the presence of residual tumor cells and may be able to predict imminent relapse in different solid tumor settings [14,15]. The measurement of cfDNA through liquid biopsy is a minimally invasive and efficient approach that can be performed repeatedly, allowing for the frequent monitoring of disease progression [16,17]. The use of liquid biopsy for HNSCC may help to provide guidance for the intensity of clinical and radiological surveillance and allow for early detection and treatment of recurrent disease [18,19].

In this review, we discuss our current understanding of HNSCC genetics and the potential role of cfDNA genomic analyses as an emerging precision diagnostic tool in the management of these patients.

## 2. Genetics of HNSCC

### 2.1. Genetic Alterations in HNSCC

Despite originating from a variety of different tissues in the upper aerodigestive tract, HNSCC cells display recurrent patterns of structural genome aberrations and acquired somatic mutations [20,21,22,23,24,25,26,27,28,29,30,31,32]. Prominent copy number variations (CNVs) include loss of 3p, 8p 9p and 17p and amplification of 3q, 5p, 8q and 11q13 [20,22,33]. The substantial instability of HNSCC genomes is also illustrated by the mean size (6.7 megabases) and mean number of CNVs which ranges between 141 and 433 per sample [20,22]. While some CNVs are shared across HNSCC (e.g., focal amplification of 3q26/28 encompassing *TP63*, *SOX2* and *PIK3CA*, or amplification of 11q13 regions encompassing *FADD* and *PPFIA1*), distinct CNV patterns reflect the HPV-associated dichotomization of HNSCC entities (Figure 1) [20,22,23]. For example, HPV+ tumors often display amplification of the genomic regions 20q11 (*E2F1*), 12p13 (*LAG3*, *TNFRSF1*) and 3q27/28 (*ATR*, *BCL6*, *PSMD2*, *MAP3K13*, *ALG3*, *IGF2BP2*), while HPV- HNSCC cells are enriched for amplifications of 7p11 (*EGFR*), 8p11 (*FGFR1*), 17q12 (*ERBB2*, *CDK12*), 5p15 (*TERT*), 9p24 (*JAK2*, *CD274*), 15q26 (*ALDH1A3*, *IGF1R*) and the coamplification of 11q13 (*CCND1*) and 11q22 (*BIRC2*, *YAP1*) [20,21,22,23,24]. In addition, HPV+ HNSCCs display deletion of regions at 11q23.3 (*BIRC2*, *BIRC3*, *ATM*), 14q32.32 (*TRAF3*), 13q14.2 (*SMAD9*, *CCNA1*) or 7q36.1 (*CDK5*, *EZH2*), while HPV- HNSCCs are commonly deleted at 9p21.3 (*CDKN2A*), 2q22.1 (*LRP1B*), 10q23.31 (*PTEN*), 9q34.3 (*NOTCH1*) and 18q21.2 (*SMAD4*) [20,21,22,23,24]. A notable HNSCC subtype beyond the HPV dichotomization is nasopharyngeal carcinoma (NPC). NPC is often associated with EBV infection and is endemic to Southeast Asia and North Africa [34,35]. NPCs show high frequencies of copy number gains at 1q, 3q, 8q 12p and 12q or deletions in 1p, 3p, 9p, 9q, 11q, 14q and 16q [34,36].

Across studies, the most commonly selected nonsynonymous single nucleotide alterations are located within the *TP53*, *FAT1*, *CDKN2A*, *PIK3CA*, *NOTCH1*, *KMT2D*, *NSD1*, *CASP8* and *AJUBA* genes [20,21,24,25,27]. Notably, many of these somatic mutations are found in the amplified regions mentioned above [20]. While the general abundance of somatic single-nucleotide mutations in HNSCC is independent of HPV status [20,37], the spectra of mutations are different in HPV + tumors [20,21,25,37,38]. The most commonly mutated gene in HNSCC is *TP53* with frequencies ranging up to 80% of HPV- cases [20,32,39,40]. Although HPV+ HNSCCs exhibit less frequent *TP53* mutations (as also observed in EBV- NPCs [34]), they usually degrade TP53 via the HPV-encoded oncogenic ubiquitin ligase E6 [41]. The TP53 tumor suppressor safeguards the functionality of many critical cellular processes such as cell cycling, DNA damage response, senescence and metabolism which cancer cells collectively hijack to acquire their neoplastic growth capabilities [42,43,44,45,46]. In the absence of activating stress signals the proteasome ensures a high turnover of MDM2-ubiquitinated TP53 [42,47]. Following activation by genotoxic and non-genotoxic stress stimuli MDM2 is rapidly inactivated leading to TP53 accumulation and post-translational modifications [42,44,47]. Activated TP53 mainly but not exclusively acts as a transcription factor [48,49]. In HNSCC, defective TP53 signaling can increase proliferation [50], promote invasiveness [51] and genomic instability [52], result in radiation resistance [53] and affect the tumor immune microenvironment [54]. As a consequence, mutations in *TP53* are associated with reduced survival [55].

After *TP53*, the second most mutated gene in HNSCC is *FAT1,* affecting around 20% of patients with HNSCC [20,56,57]. *FAT1* encodes a multifunctional type 1 transmembrane cadherin-related protein that—after proteolytic activation—acts as a modulator of oncogenic Wnt/β-catenin signaling [58], the Hippo/YAP1 signalosome [59], Ena/VASP-mediated cytoskeletal dynamics [56] and the EGFR/MAPK pathway including EGR-Hippo crosstalk [60]. FAT1 mutations are associated with invasion and metastasis [57] and response to radiotherapy [61].

Mutations in *CDKN2A*, *NOTCH1*, and *PIK3CA* are found in about 10–30% of HNSCC cases dependent on cohort size and patient selection [20,21,25,26,38,62,63]. *CDKN2A* encodes the p16^INK4A^ tumor suppressor which arrests cell cycle progression at the G1-S restriction point by inhibiting CDK4/6-mediated phosphorylation of retinoblastoma-associated protein (RB1), a prototypical cell cycle regulator and driver of carcinogenesis [64]. *CDKN2A* loss is associated with resistance to immunotherapy [65]. Interestingly, HNSCCs commonly share genetic events that cause simultaneous inactivation of TP53 and p16^INK4A^. In HPV- HNSCCs, *TP53* loss-of-function mutations are associated with *CDKN2A* point mutations or the loss of 9p21.3, which encodes *CDKN2A*, while in HPV+ HNSCCs, TP53 and RB1 are degraded by the viral E6 and E7 oncoproteins [2,20,25,38]. The *NOTCH1* tumor suppressor gene encodes a conserved transmembrane protein that exerts pivotal regulatory roles during development and substantially contributes to tissue homeostasis [66,67]. Although activating mutations have been described for, e.g., *NOTCH* [62,68] and *TP53* [38,55], it is a distinctive attribute of HNSCCs that the most commonly detected genetic events result in the inactivation of tumor suppressors or genes acting in associated pathways. This most likely reflects the fact that HNSCCs originate from basal keratinocytes of the mucosal epithelia which are capable to self-renew and to give rise to terminally differentiated epithelial cells [2,69]. Epithelial self-renewal depends, among others, on downregulation of *NOTCH*, *CDKN2A* and the action of the TP53 family member TP63 [70,71,72]. In contrast, there is no clear evidence of oncogene-driven reversal of a non-proliferative terminal differentiated epithelial phenotype [69]. The detected mutation patterns thus indicate that HNSCCs exploit these mechanisms for neoplastic transformation, especially as initiating events [2,69]. Notably, one of the few oncogenes commonly activated in HNSCC is *PIK3CA* [20,21,25,26,73,74]. Activation of PIK3CA is usually detected in advanced-stage, HPV+ tumors and accompanied by further hits within the PTEN-PI3K-Akt pathway [73,74].

In addition, classical tumor-driving gene fusions of *ALK*, *ROS* or *RET* are rarely detected in HNSCC [20]. Nevertheless, *ETV6-NTRK3* [75], *PAN3-NTRK2* [76] and *FGFR3-TACC3* [20] have been identified in a few cases.

### 2.2. Therapeutic Implications of Driver Gene Aberrations in HNSCC

Patients with HNSCC, who have failed standard first-line therapies, have limited therapeutic options and may benefit from new targeted therapies. Marret et al. ranked recurrent molecular alterations in HNSCC on the basis of the European Society for Medical Oncology (ESMO) Scale for Clinical Actionability of Molecular Targets (ESCAT) and identified six of 33 actionable alterations as the most clinically relevant: *HRAS* activating mutations, high microsatellite instability (MSI-H), high tumor mutational burden (TMB-high), *NTRK* fusions, *CDKN2A* inactivating alterations, and *EGFR* amplifications [77].

HRAS-activating mutations occur in approximately 4–8% of HNSCC patients [78]. HRAS oncogenic function is dependent on farnesylation and has been shown to be inhibited by tipifarnib, a selective inhibitor of farnesyltransferase, in *HRAS* mutant (m*HRAS*) HNSCC xenograft models [79]. In a single-arm, open-label phase II trial of tipifarnib for patients with recurrent and/or metastatic (R/M) HNSCC with m*HRAS* 20 patients were evaluable for response at the time of data cutoff [80]. The objective response rate for patients with the m*HRAS* variant allele frequency (VAF) of ≥20% was 55% (95% CI, 31.5 to 76.9), and median progression-free survival (PFS) on tipifarnib was 5.6 months (95% CI, 3.6 to 16.4) versus 3.6 months (95% CI, 1.3 to 5.2) on the last prior therapy, and the median overall survival (OS) was 15.4 months (95% CI, 7.0 to 29.7) [80]. Due to these encouraging results, the FDA has granted a “Breakthrough Therapy Designation” to tipifarnib for the treatment of patients with R/M m*HRAS* HNSCC with VAF ≥ 20% after disease progression on platinum-based chemotherapy in 2021.

In HNSCC the incidence of TMB-high, defined as ≥10 mutations per megabase (mut/Mb), is around 20% and the incidence of MSI-H is 1.2% [77]. TMB-high and MSI-H status has been correlated with the response to checkpoint blockade in basket trials which led to the tissue-agnostic FDA approval of pembrolizumab for advanced solid tumors meeting these criteria [81,82].

*NTRK* fusions are rare in HNSCC (<1%) [77]. There are currently two targeted therapeutic options for patients with *NTRK* gene fusions: the tropomyosin kinase (TRK) inhibitors entrectinib and larotrectinib [83,84]. In patients with advanced or metastatic *NTRK* fusion-positive solid tumors, the objective response rates ranged from 57% to 79% resulting in tissue-agnostic approvals by the EMA and FDA [77,83,84].

*CDKN2A* inactivating alterations that cause the hyperactivation of CDK4/6 are reported in 53.8% of HNSCC [77]. The selective CDK4/6 inhibitor palbociclib in combination with cetuximab showed promising activity in patients with platinum-resistant or cetuximab-resistant HPV-unrelated HNSCC in a non-randomized phase 2 trial [85]. However, in a double-blind randomized phase 2 trial (PALATINUS) there was no significant difference in median OS with palbociclib and cetuximab versus placebo and cetuximab [86]. Phase 2 and 3 trials are underway investigating palbociclib in biomarker selected patients with R/M HNSCC since the largest reduction in risk of death with palbociclib in the PALATINUS trial occurring in the subset with *CDKN2A* mutations [87].

*EGFR* amplifications are commonly found in patients with HNSCC [77]. Afatinib, an irreversible ERBB family blocker was evaluated as a second-line treatment in patients with R/M HNSCC in the LUX Head and Neck 1 trial [88]. Compared with methotrexate, afatinib was associated with significantly improved PFS (median 2.6 months for the afatinib group versus 1.7 months for the methotrexate group) [89]. A more pronounced benefit with afatinib was observed in patients with *EGFR*-amplified tumors [90].

In combination with chemotherapy in the first-line treatment of R/M HNSCC, *EGFR* copy number was not a predictive biomarker for the efficacy of cetuximab [91]. However, the presence of a single nucleotide polymorphism encoding EGFR-K_521_ represents an important mechanism of primary resistance to cetuximab in HNSCC [92]. This EGFR polymorphism is expressed in more than 40% of individuals and was shown to be associated with significantly shorter PFS upon palliative treatment with cetuximab plus chemotherapy or radiation [92]. *TP53* is the most frequently altered gene in HNSCC with mutations detected in over two-thirds of patients [93] but evidence-based clinical data regarding TP53 actionability are scarce [77]. *TP53* mutational status may, however, predict decreased sensitivity to cisplatin-based therapy [93]. Loss of function of p53 mutant proteins predicted a significantly lower pathologic complete response rate and suboptimal response to cisplatin-based neoadjuvant chemotherapy in patients with oral cavity squamous cell carcinoma [94].

### 2.3. Genetic Heterogeneity in HNSCC

The spatiotemporal genetic heterogeneity of solid tumors has been associated with a dismal prognosis due to decreased therapy response and higher rates of tumor recurrence [95,96]. This also applies to patients with HNSCCs, even in the setting of a favorable HPV+ tumor status [33,97,98,99].

Efforts to break the heterogeneous group of HNSCCs down into prognostically relevant subgroups utilized the bulk gene expression analysis of more than 279 tumor samples to define four distinct tumor expression subtypes [20,23,100]. Recent advances in single-cell RNA sequencing techniques have allowed for the further refinement of the HNSCC subtypes identified through bulk gene expression analysis. Specifically, these subtypes have been classified into three groups: malignant-basal, classical, and atypical tumors. Notably, the former mesenchymal subtype has been reclassified as malignant-basal tumors, which are characterized by an abundance of interspersed mesenchymal cells. While this subtyping approach has been successful in identifying distinct subgroups, there are still significant transcriptional differences observed both within and between patients. Interestingly, cells located at the leading edges of malignant-basal tumors have been found to partially express genes associated with epithelial-mesenchymal transition. This expression signature has been shown to be predictive for locoregional lymph node metastasis, highlighting the clinical significance of these subtypes [101]. However, further research is needed to fully understand the underlying mechanisms driving these transcriptional differences and how they may impact patient outcomes.

Another study has shed light on the genetic patterns of metastasis in HNSCCs, revealing two distinct subtypes with potential clinical implications. Patients with hematogenous metastasis exhibited upregulations of PD-L1 and PD-L2, suggesting that immune checkpoint inhibition may be a viable treatment option for this group. In contrast, patients with lymphatic metastasis showed a better response to chemotherapy in combination with locoregional radiotherapy [99].

In addition, prognostically relevant tumorigenic mutations were also found in tumor-adjacent tissue sometimes referred to as oral field cancerization [2,69,102]. In line with this observation, metachronous recurrent tumors were described to be concordant in only 60% of somatic nucleotide variants found in the primary tumor by whole exome sequencing [103]. The discovery of pre-malignantly transformed cell populations has important implications for understanding disease relapse and the development of secondary malignancies in HNSCC patients since these cell populations may be a source of relapse even in patients who are in full remission after first-line therapy. It could also explain the occurrence of frequently observed secondary malignancies. Future research will be needed to investigate the underlying mechanisms driving the transformation of these cells and to develop more effective treatment strategies to target them.

Taken together, liquid biopsy methods can indeed help to further elucidate the mutational landscapes of tumors, their surrounding tissue as well as their metastases. However, this complex spatiotemporal heterogeneity poses a great challenge and needs to be factored in for the future clinical application of liquid biopsy. Different techniques may need to be applied.

## 3. Genetic Analysis of Circulating cfDNA in Patients with HNSCC

### 3.1. cfDNA in Patients with Solid Tumors, Technical Challenges and Limitations

The analysis of cfDNA already generated multiple insights into tumor genetic composition [104], resistance mechanisms [105], tumor dissemination [106] and tumor evolution [107]. Due to its promising clinical applications, specifically for tumor detection, the identification of targetable driver mutations, the monitoring of disease during treatment and surveillance as well as its own prognostic relevance [108], cfDNA-based assays are increasingly being incorporated into clinical trials. This is also evident from the fact, that in February 2023 more than 1370 clinical trials were listed on clinicaltrials.gov and on euclinicaltrials.eu utilizing some form of cfDNA testing [109]. Moreover, several liquid biomarker tests received FDA approval [110]. There is a great variety of techniques ranging from fixed panels for the analysis of established tumor mutations to highly individualized approaches for the detection of patient-specific aberrations each of which has its own advantages and limitations [15,17].

The genetic heterogeneity of the primary tumor and metastatic lesions, tumor evolution on therapy or surveillance, shared genetic mutations of precursor lesions and coexisting germline mutations or clonal hematopoiesis are relevant biological challenges for all cfDNA analysis techniques [111]. Additionally, the amount of shedded tumor-derived cfDNA is very variable depending on the location, vascularization, cellular turnover and stage of the tumor among several other factors [112,113,114].

In addition to biological obstacles, the technical limitations of cfDNA analysis present significant challenges. During the pre-analytical phase, the use of specialized collection tubes with reagents for leukocyte stabilization can allow for the extension of storage and shipping times by up to 14 days at room temperature. In comparison, clinical practice often uses EDTA tubes which only offer a window of 2–4 h for further downstream processing [115,116]. Genomic deoxyribonucleic acid (gDNA) contamination is an acknowledged confounder which lowers the detection sensitivity due to interference. Controlling cfDNA input quantities is, therefore, essential to guarantee a certain sensitivity of the assay and reduce the false negative rates [117]. Commonly applied fluorometric methods for quantification are limited by the missing discrimination of cfDNA fragments and gDNA [118]. Although alternative methods such as capillary electrophoresis or quantitative polymerase chain reaction (qPCR) allow for a more precise estimation, they lack the detection of the presence of enzymatic inhibitors or again are biased by gDNA contamination. Alcaide et al. proposed a multiplex single-well droplet digital PCR assay to avoid these pitfalls [117].

Obtaining an accurate estimate of input cfDNA and minimizing gDNA contamination is crucial for precise estimations of potential tumor gene amplifications, such as HER2, which have important treatment implications. Typically, gene amplifications are deduced by calculating the relative ratio of the target gene to a copy number reference gene in close chromosomal proximity, which helps to exclude potential biases caused by genomic aneuploidy [119,120]. In summary, the optimization and assessment of analyte quality are crucial for ensuring the reliability of downstream analysis results, regardless of the technique used. It forms the foundation of the analysis process and is key to achieving accurate and consistent results.

Different techniques for cfDNA analysis have been developed, of which variants of next-generation sequencing, either amplicon- or capture-based (e.g., AmpliSeq HD [121], Safe-SeqS [122], CAPP-Seq with iDES [123] and digital droplet PCR (ddPCR) or BEAMING PCR [17]) are most often employed. These techniques are reported with limits of detections for VAFs between 0.0025% and 2% and have already been reviewed in detail elsewhere [124]. Figure 2 provides an overview of these techniques with their application in HNSCC and their sensitivity levels. All NGS-based techniques are prone to PCR errors and amplification biases depending on the library size, GC content and cfDNA fragment size. With the addition of unique molecular identifiers (UMI) cfDNA fragments can be tagged before amplification steps which enable in silico correction of these biases downstream. Computational algorithms such as iDES [123], PEC [125], TNER [126], ABEMUS [127] and SiNVICT [128] can correct for stereotypical PCR errors which become especially relevant at lower VAFs detection limits [123]. Although using different estimation models, most of these algorithms calculate and remove background mutation error counts based on healthy references.

Tumor-informed NGS-based approaches promise even further improvements in VAF detection limits; however, they are more laborious and complex due to their personalized nature and necessitate an initial tumor biopsy. Flach et al. recently described a detection limit down to 0.0006% VAF for 17 patients with HNSCC utilizing such a personalized cfDNA detection method. Thereby, the recurrence of disease could be detected 108 to 253 days before clinical progression [129]. Especially for the longitudinal tracking of patients with such sensitivities, great precautions must be taken to avoid the potential cross-contamination of samples.

ddPCR is a powerful alternative to NGS methods for the analysis of cfDNA in clinical routine due to its low cost, robustness and high sensitivity with VAFs detection of down to 0.01% [117,130]. However suitable shared mutated target genes need to be identified. Due to tumor heterogeneity and shared mutations in tumor-adjacent tissue in HNSCC described in Chapter 2.1, this might be a relevant limitation for the routine application of ddPCR.

An emerging field of cfDNA analysis is fragmentomics [131,132,133,134,135]. This term refers to applications that characterize cfDNA fragmentation and topology patterns that mirror chromatin compaction, gene regulation and the epigenome [132,135,136]. These patterns show high specificity with respect to tissue origin and disease and are thus discussed as promising cancer biomarkers [137,138]. While the general feasibility of this approach has been shown for HNSCC [133], fragmentomics is still in its infancy and many technical and conceptual issues need to be resolved.

Great improvements in cfDNA analysis have brought a variety of techniques into reach for future clinical application. Depending on the summarized limitations certain techniques might be better suited for certain clinical applications (e.g., tumor detection, profiling, surveillance). To ensure the robustness and standardization of these assays quality standards and controls are proposed which will help with the translation into clinical practice [139].

### 3.2. Specific Considerations on cfDNA in HNSCC

In HNSCC, studies examining cfDNA have not only focused on blood but also on saliva samples [16,140]. DNA that is released from the basal side of the tumor cells into the lymphatic and venous system should be detectable in the plasma, whereas DNA that is released from the apical side of the cells should be found in the saliva [141].

To explore the utility of tumor-derived DNA from different body sites for the diagnosis and surveillance of HNSCC, Wang et al. collected saliva and plasma before definite treatment for primary HNSCC (*n* = 71) and before salvage treatment for recurrent HNSCC (*n* = 22) [141]. Each tumor tissue sample was evaluated for a genetic alteration (either the presence of HPV or a somatic mutation), then this alteration was used to query the corresponding saliva and plasma samples [141]. In saliva, tumor DNA was found in 100% of patients with tumors of the oral cavity and in 47–70% of patients with cancers of other sites [141]. In plasma, tumor DNA was found in 80% of patients with oral cavity tumors, and in 86–100% of patients with cancers of other sites [141]. Thus, the sensitivity for detection of tumor DNA in the saliva was site-dependent and higher for tumors of the oral cavity [141]. Overall, increased sensitivity was demonstrated when assays of two compartments were combined [141]. Furthermore, tumor DNA in the saliva was found after surgery in three patients before the clinical diagnosis of recurrence, but in none of the five patients without recurrence [141].

Tumor-specific alterations such as gene methylation represent a strategy to differentiate between tumor-free circulating DNA and tumor-derived cfDNA in HNSCC patients [142]. Fung et al. evaluated the use of ddPCR for tumor suppressor gene methylation in the oral rinses of 50 patients with HNSCC and 58 controls for early disease detection and monitoring [143]. The degree of methylation of the markers *PAX5*, *Endothelin Receptor β (EDNRB)*, and Deleted in Colorectal Cancer (DCC) was studied in HNSCC biopsies and corresponding pretreatment oral rinses [143]. The best results were obtained for the marker *PAX5* with a sensitivity in oral rinses of 84.0% (95% CI, 70.9 to 92.8) and a specificity of 87.9% (95% CI, 76.7 to 95.0) [143]; 76.9% of the relapse cases had a rebound of methylation above presurgery levels in at least one of the tested markers before confirmed recurrence [143].

Interestingly, *PAX5* methylation analyzed by ddPCR technology was also used to assess histologically cancer-negative deep surgical margin samples obtained from 82 HNSCC surgeries by an imprinting procedure and primary tissue collection [144]. *PAX5* methylated imprint margins were an excellent predictor of poor locoregional recurrence-free survival (HR = 3.89, 95% CI, 1.19 to 17.52, *p* = 0.023) by multivariate analysis [144].

Notably, the association of HNSCC with HPV or EBV infection offers the potential to use virus-derived cfDNA as a marker. For example, it has been reported that circulating HPV DNA correlates with tumor burden [145] or staging [146]. For a detailed overview of this topic, we refer to the review of Aulakh et al. [140].

### 3.3. Liquid Biopsy in Early Stage HNSCC

A few studies have investigated minimal residual disease (MRD) detection by mutant cfDNA analysis in patients with HNSCC who underwent resection with curative intent [129,130,147,148,149]. Table 1 provides an overview of these studies.

Van Ginkel et al. show that ddPCR-based detection of *TP53* mutations in blood samples from HNSCC patients with the locoregional disease is generally feasible, opening up avenues for post-treatment surveillance [130]. Jonas et al. confirm the feasibility of ddPCR-based liquid biopsy monitoring in this setting [147]. Moreover, this group—by analysis of post-surgery blood samples—shows that mutant cfDNA identifies patients at risk for early relapse and that increasing VAF precedes clinical progression. Flach et al. show that a high-sensitivity NGS-based approach may also achieve the sensitivity necessary to detect tumor-derived mutations in cfDNA in post-surgery samples [129]. With this approach, mutant cfDNA could be detected at levels as low as a VAF of 0.0006%. In all cases of clinical recurrence, mutant cfDNA was detected prior to progression, with lead times ranging from 108 to 253 days. In a study by Egyud et al., the baseline cfDNA detection rate among seven patients with verified tumor mutations was 86% (six out of seven patients) with 68% (15/22) of the mutations detected [148]. Two of four patients with recurrent disease had detectable cfDNA prior to recurrence [148]. Longitudinal cfDNA monitoring in HNSCC patients was also performed by Kogo et al. [149]. In seven of 18 HNSCC patients who had undergone curative treatment (surgery, radiotherapy or chemoradiotherapy) cfDNA tested positive again or did not test negative, and all seven patients relapsed [149]. Patients who remained negative for cfDNA during follow-up (*n* = 11) had a significantly better prognosis than those who became cfDNA positive [149].

Despite the low patient numbers included in these trials, these studies collectively indicate that patients with post-surgical detection of mutant cfDNA eventually relapse. Liquid biopsy positivity typically precedes clinical relapse by several months.

### 3.4. Liquid Biopsy in Advanced HNSCC

Liquid biopsies have the potential to enhance precision medicine for patients with advanced HNSCC. However, only a limited number of studies have shown the effectiveness of this method for detecting druggable lesions and monitoring disease and resistance in patients with R/M HNSCC.

Galot et al. determined the utility of liquid biopsy to detect potentially actionable mutations in cfDNA [151]. They found mutant cfDNA in around 70% of patients with metastatic disease and in 30% of patients with locoregional recurrent disease by targeted NGS including some patients with potentially actionable PIK3CA mutations as well as variants not found in the matched tumor tissue. The randomized phase 2 BERIL-1 trial includes an experimental combination of buparlisib and paclitaxel applied in the second line setting in R/M HNSCC. The biomarker translational study accompanying the trial analyzed cfDNA and found that the presence of *TP53* alterations and HPV-negative status was associated with increased benefit from the combination, indicating that phosphatidylinositol 3-kinase (PI3K) inhibition may improve outcomes in this subset of patients historically characterized by poorer clinical outcome [153]. Interestingly, patients with low TMB had an improved response to buparlisib and paclitaxel as opposed to studies with checkpoint inhibitors where patients with high TMB had a better response to treatment [81,153]. Another study evaluated Bimiralisib, an inhibitor of the Phosphatidylinositol-3 Kinase pathway, in patients with R/M HNSCC after chemo- and immunotherapy [154]. Only patients with a detectable *NOTCH1* mutation in the tissue sample were included in this trial based on preclinical data supporting the susceptibility to this drug in this subset of patients. The cfDNA-based detection of *NOTCH1* mutations showed satisfactory concordance with tissue analysis, suggesting that this biomarker can be conveniently detected in the blood for future trials. However, to our knowledge, there are no data from HNSCC trials available with biomarker-guided patient selection based on cfDNA analysis. In this trial, changes in the cfDNA quantity during treatment were consistent with the clinical course and cfDNA collected at the time of disease progression showed new molecular alterations such as *PIK3CA*, *BRAF*, and *TP53* mutations [154].

In addition to pre-therapeutic screening in patients with R/M HNSCC, serial analysis of cfDNA may provide insights into tumor control and the development of resistance traits over time. Our own study investigated tumor evolution in patients with R/M HNSCC treated with cetuximab/platinum/5-fluorouracil [155]. The study used targeted NGS to detect mutations in *EGFR*, *KRAS*, *NRAS*, and *HRAS*. In patients with on-treatment progression, 46% showed acquired *RAS* mutations in cfDNA before clinical resistance emerged, indicating a significant correlation between the emergence of *RAS* mutant clones and clinical resistance. The study also showed the potential of liquid biopsies to detect imminent resistance before clinical progression occurs.

The ongoing FOCUS study (NCT05075122) investigates the combination of a cancer vaccine with pembrolizumab in patients with R/M HNSCC. The biomarker part of this study includes serial liquid biopsy monitoring in R/M HNSCC as a predictor of disease progression. To search for potentially emerging resistant tumor subclones, the liquid biopsy panel contains genes previously reported to be involved in resistance to checkpoint inhibitors.

Table 1 summarizes the presented selection of studies exploring liquid biopsy for disease monitoring in R/M HNSCC.

## 4. Conclusions

For HNSCC, there is a great need for the identification of new biomarkers due to the high risk of relapse in locoregional disease after initial treatment and the limited therapeutic options in the metastatic setting. These biomarkers should identify minimal residual disease, assess treatment response, monitor disease activity, profile genomic tumor landscape, and detect targetable alterations and resistance mechanisms. All this can be offered by the analysis of cfDNA as a new type of specific and non-invasive biomarker, detected through liquid biopsy. In HNSCC, the analysis can be performed in both the blood and saliva with increased sensitivity when both analyses are combined. However, the limitations of this method should be taken into consideration. One crucial factor in analyzing cfDNA is the technical procedure as well as the standardization of the assays. The currently available data for HNSCC show that the potential usefulness of a distinct platform and/or marker depends on the respective cancer entity, stage and also diagnostic aim. More clinical studies will be necessary to investigate whether a change in therapy, based on this cfDNA analysis, will improve patient outcomes as the primary endpoint, compared to the current standard procedure consisting of clinical and radiological measures. These studies also need to define appropriate genetic markers for specific clinical endpoints, since liquid biopsy approaches remain purely experimental at this stage. Only when these data are available, will liquid biopsy fulfill its promise as a cost-effective, minimally invasive approach for cancer diagnostics.

To conclude, cfDNA analysis is becoming increasingly important in the management of HNSCC patients in the context of personalized and precision cancer medicine. Nevertheless, despite encouraging data, further research is mandatory in order to shed more light on this analysis before it is widely integrated into daily clinical practice.

## Figures and Tables

**Figure 1 cancers-15-02051-f001:**
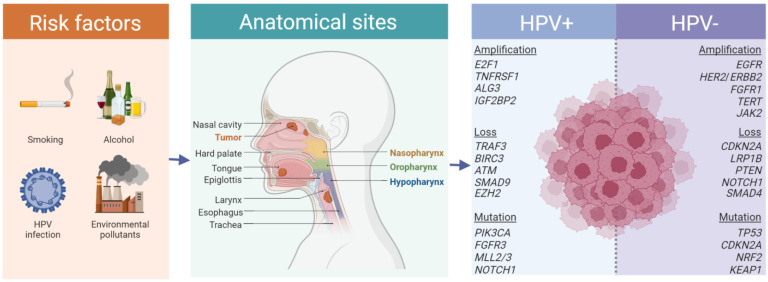
Risk factors, anatomical sites and recurrent mutations in HNSCC. HNSCC is generally subgrouped in HPV+ and HPV- subsets. Common (but not entirely exclusive) genomic alterations per subtype are indicated.

**Figure 2 cancers-15-02051-f002:**
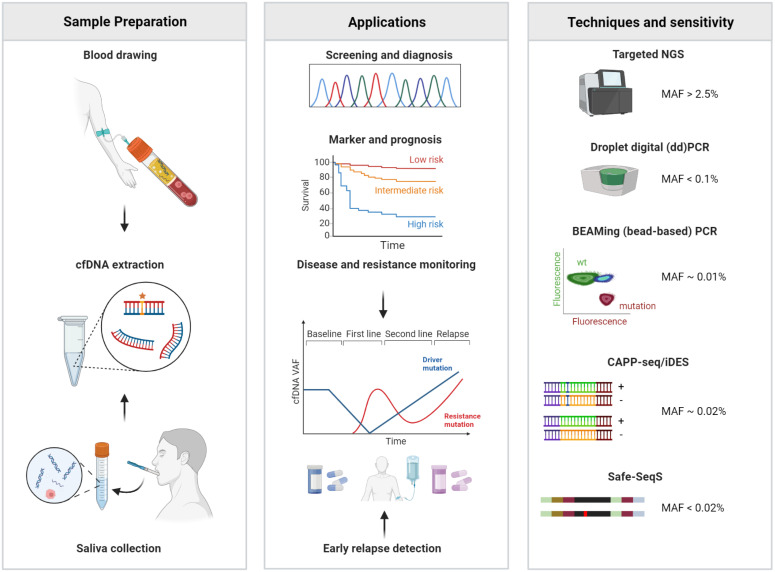
Liquid biopsy monitoring of cfDNA in HNSCC. In HNSCC, cfDNA is sampled from saliva or blood to diagnose, monitor and guide treatment decisions. Examples of technical approaches to quantify cfDNA in patient samples are indicated including thresholds of sensitivity.

**Table 1 cancers-15-02051-t001:** Key studies on liquid biopsy applications in HNSCC.

Tumor Stage and Treatment	No. of Patients (*n*)	DNA Source	Technique	Study Results	Reference
Stage II-IVA surgically treated	*n* = 6	Tumor tissue and pretreatment plasma samples	ddPCR	TP53 mutations were determined in primary tumor samples from 6 pts and in all cases pretreatment plasma samples were found positive for targeted TP53 mutations.	van Ginkel et al. [130]
Pts treated with curative intent in the IMSTAR-HN trial [150]	*n* = 19	Tumor tissue and serial plasma samples	NGS and ddPCR	11 pts were liquid biopsy positive before treatment initiation. Upon treatment, 8 of 11 pts fully cleared their ctDNA after surgery, none of these pts showed disease recurrence. 4 pts showed newly emerging or persistent ctDNA positivity in the treatment course. With a median follow-up of 93 weeks, 2 of these 4 pts had disease progression.	Jonas et al. [147]
Stage III-IVB surgically treated	*n* = 17	Tumor tissue and serial plasma samples	Whole-exome sequencing, targeted NGS	ctDNA was detected in baseline samples taken prior to surgery in 17 of 17 pts. In all cases with clinical recurrence, ctDNA was detected prior to progression with lead times ranging from 108 to 253 days.	Flach et al. [129]
Stage I-IV	*n* = 8	Tumor tissue and serial plasma samples	Whole-exome sequencing, targeted NGS	Tumor mutations were verified in 7 of 8 pts. Baseline ctDNA was positive in 6 pts. Recurrence occurred in 4 pts, 2 of whom had detectable ctDNA prior to recurrence.	Egyud et al. [148]
Treatment with curative intent	*n* = 26	Tumor tissue and serial plasma samples	dPCR	Patients who remained negative for ctDNA during follow-up after initial curative treatment (*n* = 11) had significantly better prognosis than those who reverted to ctDNA positivity (*n* = 7; *p* < 0.0001; log-rank test).	Kogo et al. [149]
R/M disease	*n* = 39	Tumor tissue and plasma samples	Targeted NGS	ctDNA was detected in 51% of pts with a higher probability of detection in metastatic than locoregional recurrent disease (70% vs. 30%, *p* = 0.025). Liquid biopsies did not reflect the complete mutational profile of the tumor but were shown to have the potential to identify actionable mutations as well as variants not found in the matched tumor tissue.	Galot et al. [151]
Pts with R/M disease treated in the BERIL-1 trial [152]	*n* = 112	Tumor tissue and plasma samples	NGS	Pts with TP53 alterations, HPV-negative status, and low mutational load derived survival benefit with the combination of buparlisib and paclitaxel.	Soulieres et al. [153]
R/M NOTCH1-mutant disease after platinum chemotherapy and PD-1-inhibitors	*n* = 6	Tumor tissue and serial plasma samples	Targeted NGS	NOTCH1 mutations in ctDNA collected at baseline were detected in 83% of pts. Changes in the ctDNA quantity during treatment were consistent with the clinical course. In addition, ctDNA samples collected at progression showed new emerging molecular alterations such as PIK3CA, BRAF, TP53, and others.	Johnson et al. [154]
Pts treated with cetuximab in curative and palliative intent	*n* = 46	Tumor tissue and peripheral blood obtained after initiation of cetuximab treatment	Targeted NGS	46% of pts with on-treatment disease progression showed acquired RAS mutations, while no RAS mutations were found in the non-progressive pts, indicating that acquisition of RAS mutant clones correlated significantly with clinical resistance.	Braig et al. [155]

R/M = recurrent or metastatic; ddPCR = Droplet Digital PCR; NGS = next generation sequencing; dPCR = digital PCR Figure 1.

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
