# Peer review of "Circulating Tumor DNA in Head and Neck Squamous Cell Carcinoma"

_cancers, 2023, doi:10.3390/cancers15072051_

Round 1

Reviewer 1 Report

Overall, this was a very nice review on ctDNA in head and neck squamous cell carcinomas. The review comprehensively discussed mutations in HNSCC, including HPV+ and HPV- groups, included a brief overview of techniques for ctDNA assessment, and finally summarized published data on ctDNA for MRD detection. 

I did feel that the ctDNA portion of the review was a little light, and did not go into as much depth as I would have liked to see on current trials specifically in HNSCC within the body of the text (as opposed to within a table). Also, what is the future of ctDNA in HNSCC, and how do the array of mutations described in the first portion of the review translate to generation of ctDNA platforms specifically tailored to HNSCC.

Author Response

Reviewer 1

Comments and Suggestions for Authors

Overall, this was a very nice review on ctDNA in head and neck squamous cell carcinomas. The review comprehensively discussed mutations in HNSCC, including HPV+ and HPV- groups, included a brief overview of techniques for ctDNA assessment, and finally summarized published data on ctDNA for MRD detection. 

I did feel that the ctDNA portion of the review was a little light, and did not go into as much depth as I would have liked to see on current trials specifically in HNSCC within the body of the text (as opposed to within a table). Also, what is the future of ctDNA in HNSCC, and how do the array of mutations described in the first portion of the review translate to generation of ctDNA platforms specifically tailored to HNSCC.

We thank the reviewer for this positive feedback and agree that the section referencing current trials should be presented in more detail in the text. We extended the respective passages. The second comment addresses the important question of implementing liquid biopsy into clinical routine. The currently available data for HNSCC shows that the potential usefulness of a distinct platform and/or marker depends on the respective cancer entity, stage and also diagnostic aim (identification of driver genes vs minimal residual disease). We now mention this point more explicitly in the section 4 (conclusions) but refrained from too much speculation.

Reviewer 2 Report

The authors describe the opportunities and challenges of using liquid biopsy for early diagnosis, monitoring, and treatment of Head and Neck Squamous cell carcinoma (HNSCC).

The review is very well written with a sufficient introduction and a brief molecular background of the disease. The genetic alterations and their therapeutic implications in HNSCC are summarized based on some of the major mutation landscape studies in the field.

The authors mentioned that the present strategies used for monitoring/diagnosis of HNSCC include regular imaging, clinical examinations, and tissue biopsy. Please also include the limitations/cost/benefits of these methods compared to liquid biopsy.

Authors have mentioned that the recurrence of HNSCC is high, a quantitative description (%) based on a literature survey would give an idea about the severity/challenges in the treatment of HNSCC.

The review also discusses the challenges posed by the genetic heterogeneity of HNSCC and technical limitations in the early detection of the relevant markers in relapse. Both figures appropriately convey the message of the description in the text. Sequencing and PCR techniques used in the sequencing and detection of cfDNA provide an adequate account of available methods for cfDNA-based studies. Authors may also want to mention the rising field of fragment-omics with applications in the same direction.

The review also highlights the challenges in the detection of low-frequency cancer-specific mutations due to PCR errors. Authors may also consider giving a few more examples of bioinformatics approaches used to suppress such errors to improve the specificity of detection.

Overall the review will be very informative for researchers and clinical practitioners in the field of HNSCC with a focus on liquid biopsy for early detection, monitoring, and treatment of the disease.

Table 1 is referenced in the main text, however, it is not provided for review. 

Author Response

Reviewer 2

Comments and Suggestions for Authors

The authors describe the opportunities and challenges of using liquid biopsy for early diagnosis, monitoring, and treatment of Head and Neck Squamous cell carcinoma (HNSCC). The review is very well written with a sufficient introduction and a brief molecular background of the disease. The genetic alterations and their therapeutic implications in HNSCC are summarized based on some of the major mutation landscape studies in the field.

The authors mentioned that the present strategies used for monitoring/diagnosis of HNSCC include regular imaging, clinical examinations, and tissue biopsy. Please also include the limitations/cost/ benefits of these methods compared to liquid biopsy.

We appreciate this positive response very much and also acknowledge the importance of considering limitations, cost and benefits of liquid biopsy. Unfortunately, we currently lack reliable study data that clearly demonstrates the usefulness of liquid biopsy for specific clinical endpoints (e.g. drug holiday, (de)-escalation studies, early relapse). As a consequence, the costs currently outweigh the benefits. We added this aspect more clearly to the limitations in section4 (conclusions) in the revised manuscript. 

Authors have mentioned that the recurrence of HNSCC is high, a quantitative description (%) based on a literature survey would give an idea about the severity/challenges in the treatment of HNSCC.

Good idea, we added this information in the text.

The review also discusses the challenges posed by the genetic heterogeneity of HNSCC and technical limitations in the early detection of the relevant markers in relapse. Both figures appropriately convey the message of the description in the text. Sequencing and PCR techniques used in the sequencing and detection of cfDNA provide an adequate account of available methods for cfDNA-based studies. Authors may also want to mention the rising field of fragment-omics with applications in the same direction.

Fragmentomics is an interesting and rapidly evolving aspect in the field of liquid biopsy. To our best knowledge, there is currently only one publication that uses a fragmentomics approach to profile cfDNA in HNSCC (Jiang et al 2020, Cancer Discovery). Nevertheless, this approach holds great promise for cancer diagnosis and monitoring. We now mention this emerging cfDNA application in the text.

The review also highlights the challenges in the detection of low-frequency cancer-specific mutations due to PCR errors. Authors may also consider giving a few more examples of bioinformatics approaches used to suppress such errors to improve the specificity of detection.

The reviewer raises a very important technical aspect of mutation profiling in PCR-based liquid biopsy approaches. In the revised manuscript, we mention a few published algorithms for detection and removal of sequencing errors. However, we refrained from providing a detailed summary since this is not within the scope of this review.

Overall the review will be very informative for researchers and clinical practitioners in the field of HNSCC with a focus on liquid biopsy for early detection, monitoring, and treatment of the disease.

Table 1 is referenced in the main text, however, it is not provided for review. 

We apologize for this problem. Since Reviewer 1 commented on Table 1, we assume that our upload was successful but there are some issues with platform. To overcome this potential problem during revision, we added Table 1 at the end of the manuscript.

Reviewer 3 Report

The authors discuss comprehensibly the current understanding of HNSCC genetics and the role of cfDNA genomic analyzes as an emerging precision diagnostic tool. Several points should be noticed as below.

1) Several reviews have been published as to this issue, for example, Aulakh SS, et al. The Promise of Circulating Tumor DNA in Head and Neck Cancer. Cancers (Basel). 2022 Jun 16;14(12):2968. What is the difference between this submitted and published similar reviews?

2) It should be noted that, EBV-associated nasopharyngeal carcinoma (NPC) belongs to the special type of head neck cancer. It should be better discuss some studies examining EBV ctDNA as a biomarker for NPC.

3) As to the nature of Cancer, one paper recently has proposed that it should be multidimensional spatiotemporal “unity of ecology and evolution” pathological ecosystem (Theranostics 2023; 13(5):1607-1631. doi:10.7150/thno.82690. https://www.thno.org/v13p1607.htm).). It should be better updated. 

Author Response

Reviewer 3

Comments and Suggestions for Authors

The authors discuss comprehensibly the current understanding of HNSCC genetics and the role of cfDNA genomic analyzes as an emerging precision diagnostic tool. Several points should be noticed as below.

We thank the reviewer for the positive feedback and will address all points in the following.

1) Several reviews have been published as to this issue, for example, Aulakh SS, et al. The Promise of Circulating Tumor DNA in Head and Neck Cancer. Cancers (Basel). 2022 Jun 16;14(12):2968. What is the difference between this submitted and published similar reviews?

The single reviews published in the mentioned special issue discuss cancer entities others than HNSCC, with often very specific focus related to therapy or resistance in a given setting. We agree that the titles of the Aulakh et al review (The Promise of Circulating Tumor DNA in Head and Neck Cancer) and our submitted manuscript (Circulating tumor DNA in head and neck squamous cell carcinoma) suggest redundancy for HNSCCs at first glance. Nevertheless, the two manuscripts emphasize different aspects of the “HNSCC-cfDNA” field. While the Aulakh et al review centers around general and technical aspects of cfDNA as a molecule and also focuses on HPV and EBV fragment detection in the context of HNSCC, our manuscript rather focuses on the mutational heterogeneity of HPV+/HPV- HNSCCs and includes more clinical aspects. We are convinced that both manuscripts complement each other well.

2) It should be noted that, EBV-associated nasopharyngeal carcinoma (NPC) belongs to the special type of head neck cancer. It should be better discuss some studies examining EBV ctDNA as a biomarker for NPC.

We agree with the reviewer that EBV-associated NPC is a noteworthy subtype of HNSCCs and that EBV (as HPV) components represent potential biomarkers of interest. We included this aspect in the revised manuscript. However, we refrained from providing too much details here, given that our manuscript focuses more on common genetic lesions shared between HNSCC subtypes and, as also discussed above in response to point 1, because Aulakh et al already reviewed this aspect.

3) As to the nature of “Cancer”, one paper recently has proposed that it should be multidimensional spatiotemporal “unity of ecology and evolution” pathological ecosystem (Theranostics 2023; 13(5):1607-1631. doi:10.7150/thno.82690. https://www.thno.org/v13p1607.htm).). It should be better updated.

We agree with the reviewer that the evolutionary and ecological aspects of cancer development are key to understand this disease. Since this review does not address cancer development, we nevertheless refrained from discussing these aspects here. However, we referenced the suggest publication in the newly added NPC section (s. response to point 2).

Round 2

Reviewer 3 Report

No other questions